# Role for Plant-Derived Antioxidants in Attenuating Cancer Cachexia

**DOI:** 10.3390/antiox11020183

**Published:** 2022-01-18

**Authors:** Wenlan Li, Kristy Swiderski, Kate T. Murphy, Gordon S. Lynch

**Affiliations:** Centre for Muscle Research, Department of Anatomy and Physiology, School of Biomedical Sciences, Faculty of Medicine, Dentistry and Health Sciences, The University of Melbourne, Melbourne, VIC 3010, Australia; wenlanl@student.unimelb.edu.au (W.L.); kristys@unimelb.edu.au (K.S.); ktmurphy@unimelb.edu.au (K.T.M.)

**Keywords:** cancer, cachexia, plant extract, antioxidants, oxidative stress, skeletal muscle, inflammation

## Abstract

Cancer cachexia is the progressive muscle wasting and weakness experienced by many cancer patients. It can compromise the response to gold standard cancer therapies, impair functional capacity and reduce overall quality of life. Cancer cachexia accounts for nearly one-third of all cancer-related deaths and has no effective treatment. The pathogenesis of cancer cachexia and its progression is multifactorial and includes increased oxidative stress derived from both the tumor and the host immune response. Antioxidants have therapeutic potential to attenuate cancer-related muscle loss, with polyphenols, a group of plant-derived antioxidants, being the most widely investigated. This review describes the potential of these plant-derived antioxidants for treating cancer cachexia.

## 1. Introduction

Cancer cachexia describes the cancer-related muscle wasting that contributes to the progression of many cancer types. It is defined as “a multifactorial syndrome exhibiting ongoing loss of skeletal muscle mass, with or without the loss of fat mass, leading to progressive muscle functional impairment” [1]. Typical clinical symptoms include anorexia, involuntary weight loss, weakness, anemia, systemic inflammation, insulin resistance and increased resting energy expenditure (REE) [2,3]. Cachexia impairs the efficacy of chemotherapy and is associated with compromised quality of life and poor survival [3]. It is estimated to account for nearly one-third of cancer-related deaths and is present in more than 50% of all cancer patients at death [4]. The distribution and severity of cachexia differs by tumor types with gastric and pancreatic cancers having the highest incidence of cachexia, and breast cancer having among the lowest incidence [3,5,6]. Despite the prevalence of cachexia and its devastating consequences, there is currently no gold standard treatment, likely because of difficulties in clinical diagnosis and the complexity of the disease. The ideal treatment for cancer cachexia is to treat the underlying cancer, but this is challenging, particularly in advanced cancers. While approximately 50 per cent of all cancer patients experience cachexia, the prevalence can be as high as 86 per cent in the last few weeks of life [7]. With such a limited therapeutic window, current management strategies focus on palliative care and improving quality of life to prolong survival. These strategies therefore aim to remove the distress of symptoms and to comfort patients and families, rather than find a cure [8]. One of the key determinants of treatment efficacy is muscle mass because of the direct link between the preservation of lean mass and patient quality of life [9]. While the pathogenesis of cancer cachexia is multifactorial, a major contributing factor is oxidative stress. While approaches to reduce oxidative stress have therapeutic potential, few studies have investigated the merit of antioxidants to treat cancer cachexia. This review synthesizes the available information regarding antioxidants with anti-cachectic properties and identifies novel candidates with potential for attenuating cancer cachexia.

## 2. Pathogenesis of Cancer Cachexia

Various factors produced by both host and tumor are thought to contribute to the development and progression of cancer cachexia (Figure 1) [10]. These multiple factors ultimately disrupt the balance between protein synthesis and protein degradation, leading to a loss of muscle mass [11,12]. Dysfunction of the membrane stabilizing dystrophin-glycoprotein complex (DGC), characterized by reduced dystrophin expression and increased glycosylation of DGC proteins, has been shown in a mouse model of gastrointestinal cancer, indicating that a loss of communication between the muscle cell membrane and the extracellular matrix could be associated with cachexia [13]. Increased metabolic stress, which occurs as a consequence of tumor burden, nutritional deficiency and various medical interventions such as chemotherapy, contributes to the development of cancer cachexia by elevating REE and exacerbating muscle loss [9]. Anti-cancer interventions such as chemotherapy can cause discomfort, nausea and anorexia in some patients, leading to reduced food intake, metabolic changes and muscle loss [14]. Since oxidative stress is a major contributor to the development of cancer cachexia, factors able to mitigate excessive oxidative stress could have therapeutic potential.

## 3. The Role of Oxidative Stress in the Etiology of Cancer Cachexia

Oxidative stress is defined as a disruption of the normal redox equilibrium in cells [15]. High levels of oxidant species, and concomitant decreased levels of antioxidant species, results in an imbalance in redox homeostasis that can accelerate the progression of tissue damage [15]. Notably, elevated levels of reactive oxygen species (ROS) and decreased antioxidant levels have been detected in the serum of tumor-bearing cachectic rats [16], implicating oxidative stress in the development of cancer cachexia [11]. Moreover, increased levels of malondialdehyde (MDA), a biomarker for oxidative stress, in gastrocnemius muscles of cachectic murine adenocarcinoma 16 (MAC16) tumor-bearing mice, supported a role for oxidative stress as an underlying mechanism of cancer cachexia [17].

Cancer-associated oxidative stress induces mitochondrial dysfunction [18], decreases protein synthesis [19], increases ubiquitin-proteasome system activity [17] and dysregulates autophagy [20]; factors all contributing to an imbalance between protein synthesis and protein degradation, and consequent muscle atrophy [15,18,20]. Increased levels of pro-cachectic cytokines, such as interleukin (IL)-1, IL-1α, IL-6 and tumor necrosis factor-α (TNF-α) are associated with weight loss, skeletal muscle catabolism and apoptosis in cancer cachexia [21,22,23]. Cancer-induced oxidative stress also elevates levels of these proinflammatory cytokines, which progress the cachexia [15]. Tumor cells can increase TNF-α expression, inducing cachexia in mice [21]. Elevated TNF-α levels promote atrophy via induction of E3 ligase genes, such as muscle RING Finger-1 (MuRF1) and muscle-specific F-box protein (atrogin-1), that mediate the ubiquitin-proteasome pathway [5]. TNF-α can also induce ROS production by the mitochondria, contributing to the redox imbalance and mitochondrial dysfunction in cancer cachexia [24]. Similarly, mitochondrial dysfunction, marked by reduced mitochondrial biogenesis and increased mitochondrial fission proteins, has been reported in an IL-6 induced mouse model of cancer cachexia, suggesting a role for IL-6 in mediating the oxidative stress in this process [18].

Clinical studies also support a role for oxidative stress as a mediator of cancer-induced muscle atrophy [25]. Levels of protein carbonyls and superoxide anion were significantly higher in muscles from cachectic lung cancer patients compared with healthy controls, indicating increased oxidative stress and proteolysis [26]. Two common characteristics in the muscles of cachectic patients are elevated levels of ROS [27] and oxidation-dependent protein modifications, which indicate the presence of an oxidative environment [15]. However, the role of oxidative stress in the pathogenesis of human cancer cachexia is less clear than in animal models since antioxidant drugs have, to date, proved less beneficial for attenuating cancer cachexia in patients [27,28].

## 4. Signaling Pathways Involved in Oxidative Stress

### 4.1. Nuclear Factor Kappa-Light-Chain-Enhancer of Activated B Cells Signaling Pathway

Nuclear factor kappa-light-chain-enhancer of activated B cells (NF-κB) signaling is a key pathway associated with oxidative stress [29]. NF-κB controls several cellular processes including inflammatory cell regulation, cell growth and apoptosis [30]. In tumor cells, NF-κB signaling has been adapted to increase cell survival and proliferation [31]. Considering the close relationship between tumor-bearing and cancer cachexia, it is likely that NF-κB also plays a role in the progression of cancer cachexia. NF-κB signaling has an important role in skeletal muscle atrophy and fat lipolysis, including post-transcriptional suppression of myoblast determination protein 1 (MyoD) mRNA inducing protein degradation, leading to muscle wasting [32]. Host and tumor-derived factors, including proteolysis inducing factor (PIF) and TNF-α, activate NF-κB signaling, enhancing proteolysis and protein degradation [33]. The promotor region of the proteasome C3 subunit provides a link to the binding site of NF-κB [34], suggesting NF-κB’s involvement in gene transcription of the ubiquitin-proteasome system. Furthermore, TNF-α treatment-induced rapid production of hydrogen peroxide and activation of NF-κB, leading to reduced total protein content and adult myosin heavy chain content in C2C12 mouse muscle myotubes in vitro [29]. Inhibition of ROS with catalase suppressed the TNF-α-induced activation of NF-κB, demonstrating a link between inflammation, oxidative stress and the regulation of muscle cell size [29]. NF-κB also mediates PIF-induced proteasome expression, activation of the ubiquitin-proteasome pathway and reduction of myofibrillar myosin levels [35], with these effects attenuated by co-administration of the NF-κB inhibitor, SN50 peptide [35]. Higher levels of ROS and activated NF-κB p65 expression have been reported in the vastus lateralis muscles of cachectic patients, together with a decreased level of activated inhibitor-κB (IκB) kinases, indicating involvement of NF-κB signaling in elevating oxidative stress in cachectic cancer patients [26]. These findings highlight the role of NF-κB activation in the pathogenesis of cancer cachexia and support the therapeutic potential for inhibition of ROS-induced NF-κB activation.

### 4.2. Nuclear Factor Erythroid 2-Related Factor 2

Nuclear factor-erythroid factor 2-related factor 2 (Nrf2) is a critical transcription factor that regulates the antioxidant response in cells under basal and stressed conditions [36] (Figure 2). Nrf2 binds to the antioxidant response element (ARE) to regulate expression of phase II antioxidant enzymes, indicating the important role of Nrf2 in maintaining body redox homeostasis [36]. Under basal conditions, kelch-like ECH-associated protein-1 (Keap1) represses Nrf2 signaling by facilitating poly-ubiquitination to promote proteasomal degradation of Nrf2 [37]. Oxidative stress from increased ROS, inflammation and/or carcinogenic factors, disrupts the interaction between Keap1 and Nrf2 to enable nuclear translocation of Nrf2, a process that activates Nrf2 signaling [38,39] to promote expression of antioxidant genes, including nicotinamide adenine dinucleotide phosphate (reduced form; NAD(P)H), quinone oxidoreductase-1 (NQO1), glutathione-S-transferase (GST), superoxide dismutase (SOD) and hemeoxygenase-1 (HMOX1), to attenuate the oxidative burden [36,40].

Activation of Nrf2 signaling also elicits a strong cytoprotective response to impair the progression of carcinogenesis [37]. While Nrf2 has protective effects against various cancer types [44,45], some studies have suggested Nrf2 activation can have negative effects by elevating the antioxidant response that contributes to a resistance to cancer treatments [46,47,48]. Moreover, Nrf2 signaling has been linked to the initiation and progression of lung cancer in vivo [49]. Nrf2-deletion in mice increased lung tumor growth but reduced the number of tumors with cancer progression [49]. These findings suggest a dual role of Nrf2 in cancer as it prevents tumor initiation but facilitates malignant tumor progression. A delicate balance therefore exists between ROS levels and antioxidant content, indicating the need for precision when treating cancer or cancer-related muscle wasting with antioxidants.

## 5. Polyphenols to Reduce Oxidative Stress in Cancer Cachexia

Despite oxidative stress being a main driver of cancer cachexia, there is a dearth of studies investigating the therapeutic potential of antioxidants for treating cancer-related muscle wasting (Table 1). Polyphenols are common phytochemicals with protective effects against oxidative stress-related diseases. They are widely distributed across plants and can be found in various plant-based foods, including fruits, vegetables and whole grains with rich antioxidant sources [50]. Due to their antioxidative and anti-inflammatory properties, the potential benefits of polyphenols against different cancers have been investigated extensively [51]. The potential for polyphenols to attenuate cancer cachexia has been attributed to a possible preservation of muscle mass through inhibition of NF-κB signaling [50].

### 5.1. Epigallocatechin-3-Gallate

Epigallocatechin-3-gallate (EGCG) is the dominant antioxidant in green and black tea extract, having antioxidant, anti-inflammatory and anti-cancer functions [34]. EGCG suppresses cancer cell proliferation, inducing cell apoptosis and inhibiting cell invasion and migration to attenuate tumor progression [52]. In cell culture, 48 h EGCG treatment suppressed the growth of Lewis lung carcinoma (LLC) cancer cells in a dose-dependent manner, confirming its anti-tumor effects [34]. In the LLC tumor-bearing mouse, 12 days of EGCG treatment reduced tumor mass and volume and attenuated the loss of body weight without altering anorexia [34]. The attenuation of muscle wasting was attributed to inhibition of NF-κB and the downstream E3 ligases, MuRF1 and atrogin-1. Furthermore, ECGC treatment decreased leukocytic infiltration, thus reducing inflammation in skeletal muscles of tumor-bearing mice [34]. In addition to the suppression of tumor growth, high dose EGCG treatment reduced the survival rate of a healthy, baby hamster kidney cells (BHK-21) by 50%, [34], suggesting that higher concentrations of EGCG cause cytotoxicity in normal cells and interrupt healthy cell growth. Given that EGCG treatment in the LLC-injected mice was administrated several days prior to tumor palpability, it remains to be determined whether the protective effect of EGCG on muscle mass was due to its anti-cancer effects or direct effects on the muscle. Moreover, EGCG has a low systemic bioavailability [53], which may reduce the efficacy of EGCG treatment in clinical studies and in vivo animal studies. EGCG treatment for cancer cachexia is a relatively preliminary concept in the field and further investigation is needed to confirm therapeutic potential.

### 5.2. Resveratrol

Resveratrol is found most abundantly in the skin of grapes, peanuts and pine bark, and has been shown to exert anti-cancer effects in vitro and in vivo [54]. In Yoshida ascites hepatoma (AH-130) cells in vitro and in rats implanted with AH-130 cells, resveratrol administration reduced tumor cell number via induction of AH-130 cell apoptosis [54]. In Ehrlich ascitic carcinoma bearing mice, the combined treatment of resveratrol (10 mg/kg) with the chemotherapeutic drug, doxorubicin (5 mg/kg), twice a week via intraperitoneal injection, exerted the best reduction in tumor size and prolonged survival compared with either treatment alone [55]. These findings demonstrate the potential for resveratrol to enhance the anti-cancer effect of chemotherapies by decreasing inflammation and the oxidative stress associated with chemotherapy [55]. 

**Table 1 antioxidants-11-00183-t001:** Effect of treatment with polyphenols for cancer cachexia.

Types	Experimental Setting	Treatments	Findings	References
EGCG	In vivo 6–8-week-old male LLC-tumor-bearing mice (C57BL/6)	Low dose (0.2 mg/kg/day), high dose (0.6 mg/kg/day) via oral gavage;	↓ NF-κB	[34]
			↓ NF-κB-mediated ubiquitin– proteasome proteolysis	
		12 days pre-treatment or 30 days post-tumor treatment	↓ atrogin-1 and MuRF1 expression	
			↓ tumor-induced muscle atrophy	
Resveratrol	In vivo 6–10-week-old female C-26 tumor-bearing mice (CD2F1)	200 mg/kg/day via oral gavage for 11 days	↓ NF-κB	[56]
			↓ atrogin-1 and MuRF1 expression	
			↓ tumor-induced muscle atrophy	
			No effect on tumor growth	
	In vivo 5-week-old male Wistar AH-130 tumor-bearing rats	1 mg/kg/day via intraperitoneal (i.p.) injection to AH-130 tumor bearing rats for 7 days	No effect on skeletal muscle and whole body mass	[57]
	12-week-old male LLC-tumor-bearing mice (C57BL/6)	5 or 25 mg/kg/day via i.p. injection to LLC-tumor bearing mice for 15 days	Failed to attenuate cancer cachexia in different tumor-bearing rodents	
	In vivo 10-week-old female BALB/c mice	20 mg/kg/day via i.p. injection for 15 days	↓ muscle wasting	[58]
			↑ gastrocnemius and soleus muscle mass	
			↓ tumor growth	
			↑ limb strength gain	
			↑ muscle fiber (I & II) cross-sectional area, ↓ muscle abnormalities	
			↑ sirtuin-1 protein expression	
			↓ atrogin-1 and MuRF1 expression	
			↓ forkhead box O3 (FoxO3)	
			↓ signaling markers NF-κB and p50	
Curcumin	In vivo 10-week-old female LP07 tumor-bearing BALB/c mice	1 mg/kg/day via i.p. injection for 15 days	↓ muscle wasting	[58]
			↑ gastrocnemius and soleus muscle mass	
			↑ limb strength gain	
			No effect on tumor growth	
			↑ muscle fiber (I & II) cross-sectional area, ↓ muscle abnormalities	
			↑ sirtuin-1 protein expression	
			↓ atrogin-1 and MuRF1 expression	
			↓ FoxO3	
			↓ signaling markers NF-κB and p50	
	In vivo MAC16-colon tumor-bearing mice	Low dose (100 mg/kg/day), high dose (250 mg/kg/day) via oral gavage for 20 days	↓ muscle wasting with low dosage	[28]
			↑ body weight, muscle hypertrophy with high dosage	
			↓ proteasome complex activity	
			Inhibited NF-κB pathway	
	In vivo Male Wistar AH-130 tumor-bearing rats	20 μg/kg body weight via i.p. injection for 6 days	↓ tumor growth	[59]
			Failed to attenuate cancer cachexia	
Carnosol	In vitro C2C12 myotube	3.125 μM to 25 μM concentration of carnosol incubated with C-26 cancer medium for 48 h in C2C12 myotubes;	In vitro: High dose (25 μM) had no toxic effect to C2C12 myotubes;	[60]
			↓ C-26 tumor-induced muscle wasting in C2C12 myotubes in dose-dependent manner	
			↑ MyoD, p-Akt at high dose of carnosol	
			↓ MuRF1, p-p65/p65 at high dose of carnosol	
	In vivo 6–8-week-old male C-26 tumor-bearing, BALB/c mice	10 mg/kg/day via i.p. injection from the day after tumor injection for 16 days	In vivo: ↑ body weight	
			No effect on tumor growth	
			↑ MyoD, myosin heavy chain	
			↓ p-p65/p65 ratio	
Quercetin	In vivo 15-week-old Apc^Min/+^mice	25 mg/kg/day via oral gavage for 3 weeks	Attenuated ↓ body mass	[61]
			↑ gastrocnemius and quadriceps muscle mass	
			No change in soleus muscle mass	
			No improvement in muscle function	
			↓ plasma IL-6	
	In vivo 9-week-old C-26 tumor-bearing male CD2F1 mice	250 mg/kg added to daily chow diet for 20 days	↑ body weight	[62]
			↑ food intake	
			No change grip strength	
			Prevented tumor-induced ↓ muscle volume	
			No change in tumor weight	
			↑ gastrocnemius and tibialis anterior muscle mass	
Rutin	In vivo 6-week-old K14-HPV16 mice	413 mg/kg/day to daily diet for 24 weeks	↑ survival	[63]
			No change in body weight	
			↑ gastrocnemius muscle weight	
			↓ NF-κB signaling pathway	
Genistein and daidzein	In vivo 8-week-old male C57BL/6 mice with LLC tumors	Normal diet mixed with 40.74% of soyaflavone HG (containing high genistein and daidzein contents) for 3 weeks	No change in food intake or body mass	[64]
			↑ gastrocnemius muscle weight and myofiber size	
			No change in tumor mass	
			No change in plasma IL-6 or TNF-α	
			↓ atrogin-1 and MuRF1 expression	
			↓ phosphorylation of extracellular signal-regulated kinase (ERK)	
Morin	In vitro LLC cells and C2C12 myotubes	In vitro: 10, 50, 100, 200 μM treated to LLC cells and C2C12 myotubes for 48 h	In vitro: ↓ cell viability of LLC cells with 100 and 200 μM	[65]
			↑ cell viability of C2C12 myotubes with 10 μM; no cell death at high dose (100 and 200 μM)	
			↓ protein synthesis shown in LLC cells using SUnSET method; no significant changes were found with C2C12 myotubes.	
	In vivo 6-week-old male C57BL/6 mice with LLC tumors	In vivo: Morin-rich (0.1% *w/w*) diet for 3 weeks	In vivo: Attenuated↓muscle mass and gastrocnemius muscle myofiber size	
			↓ tumor mass	

↓—decreased and ↑—increased.

Resveratrol has been studied as an anti-cachectic treatment to attenuate muscle atrophy. Daily administration of resveratrol (200 mg/kg/day) via oral gavage attenuated the loss of lean body and fat mass, and gastrocnemius muscle mass in C-26 tumor-bearing mice [56]. These effects were associated with inhibition of IκB kinase, the activator of NF-κB signaling, to prevent nuclear translocation and accumulation of NF-κB [56] and subsequently inhibiting expression of the downstream E3 ligases, atrogin-1 and MuRF1 [56]. Resveratrol inhibited PIF-induced activation of NF-κB in MAC16 tumor-bearing mice and attenuated the loss of muscle mass and whole body mass [35]. Moreover, resveratrol supplementation prevented TNF-α-induced myotube atrophy via activation of the Akt/mammalian target of rapamycin (mTOR)/FoxO1 signaling pathway, evident from increased Akt, ribosomal protein S6 kinase beta-1 (p70S6K), mTOR and eukaryotic translation initiation factor 4E-binding protein 1 (4E-BP1) phosphorylation and decreased FoxO1 protein expression [66]. These findings suggest that the mechanisms underpinning the protective effect of resveratrol for cancer cachexia likely involve a restoration of the balance between protein synthesis and protein degradation.

In mice injected with LP07 adenocarcinoma cells, the anti-cachectic effects of resveratrol (20 mg/kg/day for 15 days via intraperitoneal injection) were linked to activation of sirtuin-1 and attenuation of FoxO3 signaling [58]. Sirtuin-1 is a histone deacetylase enzyme that helps maintain muscle mitochondrial content and function [67]. The resveratrol-dependent activation of sirtuin-1 is proposed to improve mitochondrial biogenesis to alleviate oxidative stress in cachectic LP07 tumor-bearing mice [58]. FoxO3 is involved in the skeletal muscle ubiquitin-proteasome, autophagy-lysosomal and mitochondrial autophagy pathways [68]. However, resveratrol treatment also inhibited tumor growth, making it difficult to discern whether the attenuation of muscle wasting was due solely to a reduced tumor burden [58]. While these studies have shown promising effects of resveratrol, findings from other studies question the benefit of resveratrol for treating cancer cachexia. Intraperitoneal injection of resveratrol failed to attenuate the loss of muscle mass and body mass in LLC-tumor bearing mice (at both 5 mg/kg/day and 25 mg/kg/day for 15 days) and exacerbated the reduction in food intake and loss of gastrocnemius, heart and white adipose tissue mass in AH-130-tumor bearing rats (1 mg/kg/day for seven days) [57]. The discrepancies in the findings between the studies may be attributed to differences in animal models and/or tumor type [56,57]. Further studies are required to resolve these conflicts and determine the mechanisms underlying the therapeutic potential of resveratrol for cancer cachexia.

### 5.3. Curcumin

Curcumin, a component of turmeric, has been investigated as a potential treatment for cancer cachexia [69] due to its anti-inflammatory, antioxidative and anticarcinogenic functions as a nutritional supplement [28,59]. Curcumin is a nontoxic phytochemical with demonstrated potential to attenuate tumor growth in preclinical and clinical studies via suppression of NF-κB activity [59,70]. In MAC16 tumor-bearing mice, curcumin prevented muscle wasting and reversed existing muscle loss [28]. Treatment with curcumin c3 complex has been used in human clinical trials [71], where it protected skeletal muscle from wasting when orally administered a low dose (100 mg/kg/day) for 20 days, and induced weight gain relative to the control tumor-bearing group when administered at a higher dose (250 mg/kg/day) [28]. In MAC16 colon tumor-bearing mice, curcumin treatment attenuated the PIF-induced increase of the 20S proteasome, and decreased expression of NF-κB, atrogin-1 and MuRF1, indicating that protection from muscle wasting was attributed to suppression of the ubiquitin-proteasome pathway and subsequent protein degradation [28]. It should be noted that MAC16 colon tumor-bearing mice exhibit a gradual loss of body mass and muscle mass over 21 days, enabling the efficacy of curcumin supplementation to protect skeletal muscle mass to be assessed over a longer period [28]. Curcumin was also shown to attenuate loss of body mass and improve muscle mass and limb strength gain in cachectic LP07 tumor-bearing mice without altering tumor size [58]. These effects were associated with increased cross-sectional area (CSA) of type I and type II muscle fibers, and a reduced proportion of muscle fibers with internal nuclei and inflammatory cell infiltration, in gastrocnemius and soleus muscles [58]. Compared to studies reporting beneficial outcomes of curcumin, administration (20 µg/kg/day for six days via intraperitoneal injection) failed to improve the cachectic pathology in AH-130 tumor-bearing rats despite having anti-tumor effects [59]. The disparity in these reports may arise from differences in dose, route of administration and treatment duration in different animal models, as well as the low systemic bioavailability of curcumin [72].

While studies investigating curcumin efficacy in cancer cachexia did not measure oxidative stress directly, the role of curcumin for attenuating oxidative stress in skeletal muscle is well established. Curcumin reduced exercise-induced oxidative stress, evidenced by decreased levels of serum lactate and muscle MDA [73]. Oral curcumin treatment (100 mg/kg/day) for 14 days reduced hypobaric hypoxia-induced oxidative stress and increased muscle fiber number in Sprague Dawley rats [74]. The antioxidative effect of curcumin was linked with reduced activity of NF-κB and activation of Nrf2 signaling [73], effects that reflect regulation of redox balancing, protein synthesis and protein degradation. Thus, regulating redox balance may be one mechanism by which curcumin attenuates cancer cachexia.

The therapeutic potential of curcumin has also been explored in clinical trials. Based on the limited data available, curcumin reduced expression of NF-κB in some patients with pancreatic cancer [70]. Due to its poor oral absorption and weak bioavailability, curcumin supplementation had only limited benefit for these pancreatic cancer patients [70]. However, the difficulty in accurately determining redox status in skeletal muscle [75], may explain why evaluating the efficacy of antioxidant supplements like curcumin has proved challenging, especially when assessments in trials rely on measures of antioxidant levels in the blood. Improvements in the accuracy of these outcome measures are required to better evaluate the therapeutic potential of curcumin for cancer cachexia.

### 5.4. Carnosol

Carnosol is a bioactive diterpene compound present in rosemary, with antioxidant, anti-inflammatory and anti-cancer properties [60]. The antioxidative function of carnosol has been well characterized and includes protection from lipid peroxidation [76], suppression of nitric oxide production and gene expression of inducible nitric oxide synthase [77] and amelioration of the damage caused by UVB-induced ROS [78]. Carnosol can inhibit the activities of NF-κB signaling to protect against free radical damage [77] and it has been shown to reduce tumor growth in mouse models of intestinal cancer [79], breast cancer [80] and skin cancer [78].

Carnosol can protect against cancer-induced muscle wasting in in vitro and in vivo models [60]. Carnosol supplementation attenuated C2C12 myotube atrophy after exposure to C-26 cell conditioned media and ameliorated the loss of body mass in C-26 tumor-bearing mice [60]. These effects were associated with downregulation of MuRF1 expression and upregulation of Akt phosphorylation and MyoD expression, indicating suppressed protein degradation and increased protein synthesis [60]. Both in vitro and in vivo models showed reduced phosphorylation of p65, implicating a role for carnosol in suppressing NF-κB signaling in cancer cachexia [60]. While carnosol had a protective role in C-26 tumor-bearing mice by maintaining body mass and adipose tissue mass, skeletal muscle mass was not improved. This suggested the increase in body mass resulted from an attenuation of fat lipolysis rather than direct effects on the regulation of skeletal muscle mass [60]. These interesting findings warrant further investigation of the therapeutic potential of carnosol for cancer cachexia. Moreover, a recent study revealed the synergistic effect of carnosol and a chemotherapeutic drug, cisplatin, where combined therapy induced the highest rates of apoptosis in MCF-7 and MDA-MB-231 breast cancer cell lines [81]. Therefore, carnosol has potential to become part of a combination therapy in the treatment of cancer and cancer-induced muscle wasting.

### 5.5. Quercetin and Rutin

Quercetin is an abundant flavonoid and prominent dietary antioxidant in various fruits and vegetables, such as onions, tomatoes and apples [82]. Quercetin has antioxidative and anti-inflammatory functions, and hence therapeutic potential for treating cancer [83,84]. Supplementation with quercetin (0.05% (*w/w*) in food) for nine weeks protected against TNF-α-induced skeletal muscle atrophy via activation of Nrf-2 signaling and inactivation of the NF-κB signaling pathway to overcome oxidative stress in the C57BL/6 mouse model of high fat diet-induced obesity, confirming the antioxidant and anti-inflammatory properties of quercetin [82]. Quercetin has significant bioavailability compared with other polyphenols, with detection in plasma 12 h after oral intake [62]. The high absorption in plasma was associated with whole body accumulation of quercetin since it was detected in different tissues such as the liver, skeletal muscle, heart and brain, resulting in the slow clearance of quercetin metabolites from the body [62].

Quercetin has been shown to protect against cancer-induced muscle wasting in vivo [61,62]. Oral administration of quercetin (25 mg/kg/day) to Apc^Min/+^ mice for three weeks attenuated the loss of whole body mass and increased gastrocnemius and quadriceps muscle mass [61]. These effects may have resulted from reduced inflammation, based on the decrease in plasma IL-6 levels [61]. However, treatment failed to improve muscle function in the Apc^Min/+^ mouse model of cancer cachexia [61]. Supplementation of 250 mg/kg quercetin to a daily chow diet for 20 days attenuated both the loss of body mass and the reduction in gastrocnemius and tibialis anterior muscle mass in C-26 tumor bearing mice, but did not improve grip strength [62]. Micro-CT analysis of the hindlimb revealed that quercetin supplementation completely prevented the tumor-induced reduction in muscle volume [62]. A substantial (albeit a non-statistically significant) decrease was detected in the expression of E3 ubiquitin ligases, atrogin-1 and MuRF1, in treated mice, indicating an attenuation of protein degradation. While these studies suggest a positive outcome of quercetin with a potential anti-cachectic function, the underlying mechanism remains undetermined. Interestingly, tumor mass was not significantly decreased with quercetin. Furthermore, the experiment did not control for the increase in food intake in the quercetin supplemented group [62]. While quercetin has proposed anti-cachectic benefits, a mechanism for these effects has not been established at the cellular level, and this diminishes the significance of these outcomes. Moreover, oral quercetin (50 mg/kg/day) for 1 h prior and during 15-day doxorubicin exposure, reduced chemotherapy-induced oxidative stress in the spleen via suppression of apoptosis, reducing inflammation and increasing the antioxidant response in Sprague–Dawley rats [85]. The protective effect of quercetin against chemotherapy-induced cytotoxicity indicates its potential for attenuating chemotherapy associated muscle atrophy in cancer cachexia. Future studies using pair-fed groups to control for potential treatment-related changes in food intake, as well as additional clinically relevant end-point analyses such as survival, are warranted in order to fully investigate the therapeutic potential of quercetin in cancer cachexia.

Similar to quercetin, rutin, a quercetin glycoside more commonly seen in edible plants, has similar high bioavailability, making it a suitable candidate for nutritional therapy [86]. Supplementation with rutin (413 mg/kg/day) for 24 weeks increased survival in HPV16-tumor bearing mice and increased gastrocnemius muscle mass, which was associated with inhibition of NF-κB signaling [63]. Rutin also alleviated carcinogenesis via suppression of cyclo-oxygenase-2 in K14-HPV16 mice [87]. Studies have demonstrated the synergistic benefit of rutin in combination with chemotherapeutic drugs to further reduce cell proliferation in different cancer cell lines via activation of apoptosis, thereby enhancing the anti-cancer effect of chemotherapy [88]. Thus, rutin has significant potential in combination therapies to attenuate cancer cachexia.

### 5.6. Genistein, Daidzein and Morin

Genistein and daidzein are isoflavones abundant in soy products, with anti-inflammatory and antioxidative properties [64]. Supplementation with soy isoflavones (mainly genistein and daidzein) for three weeks attenuated LLC tumor-induced muscle wasting by increasing both the overall muscle mass and size of individual muscle fibers within the gastrocnemius. These effects were associated with decreased expression of the ubiquitin-related E3 ligases, atrogin-1 and MuRF1, and likely mediated by ERK signaling to exert muscle-protecting function against cancer cachexia [64]. Soy isoflavones have also been widely discussed in the context of breast cancer, since they are plant-derived substances that activate signaling via estrogen receptors [89]. However, the benefit of soy isoflavones for breast cancer patients has been controversial [90]. In the MCF-7 breast cancer cell line, high dose (100 μM) of genistein combined with the chemotherapeutic agent, cisplatin, suppressed breast cancer cell growth and proliferation, whereas 10 μM of genistein antagonized the action of cisplatin to induce cancer cell apoptosis [89]. Further investigation confirmed that oral genistein supplementation (5 mg/kg/day) for three weeks counteracted cisplatin chemotherapy in breast cancer-bearing mice [91]. The anti-cancer effect of these soy isoflavones (genistein in particular) therefore remains controversial due to insufficient evidence. Such conflict raises doubt about the potential of these soy isoflavones to attenuate cancer-induced muscle atrophy. Future investigation is warranted to address these concerns.

Morin is a type of flavonoid, found in plants including Moraceae, Malpighiaceae, Myrtaceae, almond hulls and seaweeds [65]. Supplementation with a morin-rich diet for three weeks reduced tumor weight and progression in LLC-tumor bearing mice, demonstrating a powerful anti-cancer effect [65]. In addition, morin has been shown to have anti-cachectic potential, by attenuating cancer-induced muscle wasting in these same mice [65]. In vitro studies revealed that morin suppressed cancer cell growth via decreasing protein synthesis [65]. In contrast, morin (10 μM) increased protein synthesis in C2C12 myotubes, and this was associated with increased cell viability [65]. Moreover, oral morin treatment (50 mg/kg/day) for 30 days reduced the damage caused by a single injection of cisplatin in Sprague–Dawley rats by ameliorating chemotherapeutic drug-induced oxidative stress and activating antioxidant signaling cascades in isolated renal mitochondria [92]. These findings suggest a benefit for morin supplementation in reducing chemotherapy-induced cytotoxicity, which may also have positive effect on ameliorating chemotherapy-related oxidative stress in skeletal muscle. Further investigation is needed to evaluate the potential for morin to attenuate cancer cachexia.

## 6. Other Antioxidants with Therapeutic Potential for Cancer Cachexia

Given the etiology of cancer cachexia is multifactorial, it is likely that only targeting a single factor may be inadequate and that a multimodal approach is needed to confer benefits. Identifying the most effective strategies that improve food intake, enhance protein synthesis, reduce protein degradation and decrease inflammation, are needed before testing the safety and efficacy of their different combinations. Since oxidative stress is one of the main contributing factors favoring protein breakdown over synthesis, through mitochondrial dysfunction and dysregulation of autophagy, including a safe and effective antioxidant could be an important for combination therapies. To date, all of the antioxidants examined for treating cancer cachexia are pharmacologically similar in their anti-inflammatory, anti-cancer and antioxidative properties. Thus, it is important to also investigate other antioxidants with similar pharmacological properties to increase the feasibility and efficacy of combination approaches.

### 6.1. Ursolic Acid

Ursolic acid (UA) is a natural pentacyclic triterpenoid carboxylic acid commonly found in some herbs and fruit wax, such as apples, pears, prunes and other fruits [93], which has antioxidant, anti-inflammatory and anticancer properties [94]. As a plant-derived antioxidant, the anti-cancer effect of UA has been well characterized, with anti-inflammatory and chemoprotective effects demonstrated in various cancer types in cell (in vitro) and rodent (in vivo) models [93]. The anti-cancer and anti-inflammatory effects of UA are mediated via various signaling cascades, including the signal transducer and activator of transcription 3 (STAT3) and NF-κB signaling pathways [95] and the Akt/p70S6K signaling pathways [96] to suppress tumor growth, survival and metastases [93].

In addition to promising anti-cancer properties, UA was shown to improve skeletal muscle health in various pathologies (Table 2). Acute (12 h) and short-term (three days) supplementation with UA attenuated muscle wasting induced by fasting [97] and hypobaric hypoxia, by promoting protein synthesis [98], attenuating protein degradation [97] and reducing oxidative stress [98]. Longer UA treatment (5–6 weeks) improved muscle mass and function and increased slow and fast muscle fiber size without altering fiber type composition [97,99]. Moreover, oral UA supplementation (0.27% in food) for five weeks caused muscle hypertrophy and improved muscle function in healthy mice [97]. 

Oral UA supplementation (20 mg/kg/day) for three days to healthy rats increased body mass and muscle protein content, in the absence of changes in pro-inflammatory cytokines, ROS levels or oxidation markers [98]. While the beneficial effects of UA on muscle have been established, higher doses of UA (200 mg/kg, twice daily via intraperitoneal injection) for seven days reduced body mass and cellular energy stores in aging, despite improved muscle regeneration and a fast-to-oxidative fiber type shift [101,102]. These discrepancies may be due to differences in methodology across studies, including treatment duration and route of administration and highlight the need for clarifying studies (Table 2).

The attenuation of atrophy by UA is proposed to occur via induction of insulin-like growth factor-1 (IGF-1) and activation of Akt/mTOR1 signaling [97,100]. UA-mediated IGF-1 induction is localized to skeletal muscle, increasing muscle mass via Akt/mTOR signaling [99], evident from enhanced Akt and p70S6K phosphorylation and reduced glycogen synthase kinase 3β (GSK3β) expression [98,100]. While short-term UA treatment was insufficient to increase muscle IGF-1 content, it was able to enhance resistance exercise-induced activation of Akt/mTOR signaling [100]. Longer-term UA treatment would be expected to increase IGF-1 levels and Akt phosphorylation, whereas acute UA treatment would likely to enhance Akt/mTOR signaling cascades. In clinical studies, UA demonstrated potential for attenuating resistance-exercise-induced muscle damage in a pilot study [94], therefore supporting the therapeutic potential for UA to attenuate muscle wasting in cancer. The high bioavailability of UA [93] and its ability to sustain upregulated protein synthesis, make it a promising candidate for combination therapies.

### 6.2. Sulforaphane

Sulforaphane (SFN) is a natural dietary isothiocyanate that was first identified and isolated from broccoli in 1992 [103]. It is produced by the enzymatic action of myrosinase on glucopharanin, a relatively stable precursor found in cruciferous vegetables such as broccoli, Brussels sprouts and cabbage [104,105]. SFN is a potent inducer of detoxification enzymes with both chemoprotective and anti-carcinogenic potential [103]. It induces several phase II enzymes including GST and glucuronosyltransferases, defining its antioxidative action [104]. The expression of these enzymes continues throughout the duration of SFN administration and has been reported in various organs, indicating its systemic antioxidant response [36]. In addition, SFN has an absolute bioavailability of around 80%, which is much higher than that of other polyphenols commonly used in dietary supplements [106,107]. Similar to curcumin, SFN is a major regulator of Nrf2 signaling [108], enhancing nuclear translocation of Nrf2, where it binds the ARE to produce phase II antioxidant enzymes [38,39]. SFN therefore plays an important role in the survival signaling pathway in the face of extensive oxidative stress resulting from cancer, muscle atrophy and chemotherapy [37,40,109]. Overall, SFN shows promising potential as a bioactive compound for treating toxicity related to oxidative stress, although further studies need to confirm its therapeutic potential.

SFN’s ability to combat carcinogenesis was first proposed in the 1990s [104] and it has more recently gained attention as a potential dietary supplement for treating cancer [106]. The chemoprotective effects of SFN are well described, with potential to treat cancers of the breast, prostate and lungs [38,110,111,112]. SFN suppressed KPL-1 human breast cancer cell proliferation and induced apoptosis in a dose-dependent manner in in vitro and in vivo studies [110], and restored Nrf2 expression and downstream antioxidative enzymes in TRAMP prostate tumor-bearing mice [112,113]. While preliminary clinical data support the potential for SFN to treat prostate cancer [114], further clinical trials with larger cohorts are required to more comprehensively determine SFN’s therapeutic efficacy.

SFN has reported benefits for skeletal muscle atrophy (Table 3). SFN attenuated dexamethasone-induced C2C12 myotube atrophy in vitro by inhibiting the increase in myostatin and atrogin-1 expression [40]. Similarly, in porcine muscle stem cells (satellite cells), SFN inhibited myostatin gene expression while promoting an increase in mRNA expression of myostatin inhibitors, Smad7 and Smad-specific E3 ubiquitin protein ligase 1 (Smurf1) [115]. These findings reveal a role for SFN in decreasing the rate of protein degradation via reducing E3 ligase-mediated ubiquitination in muscle stem cells and in differentiated myotubes. Furthermore, acute SFN pre-treatment increased muscle cell survival rate and myotube diameter and reduced intracellular ROS levels under menadione-induced and starvation-induced oxidative stress [39,116], suggesting SFN can attenuate oxidative stress in C2C12 myotubes.

SFN administration has also been shown to influence the regulation of skeletal muscle growth and regeneration [40,115,117]. In C2C12 myoblasts, 4 h exposure to SFN induced Nrf2 activation, but inhibited or delayed myotube differentiation, evidenced by a decrease in MyoD, myogenin and myosin expression [117]. In addition, siRNA-mediated reduction of Nrf2 revealed an inverse association between Nrf2 activation and the progression of myotube differentiation in C2C12 muscle cells [117]. These findings are consistent with the alternate role of Nrf2 by inhibiting the initiation of cell growth [45]. Similarly, administration of SFN to isolated muscle stem cells reduced MyoD mRNA expression and proliferation [115]. There is considerable variability between studies investigating the potential benefits of SFN treatment on skeletal muscle in vitro because of methodological differences, including different SFN concentrations and treatment durations. While SFN has been shown to attenuate chemical-induced muscle damage [40,116,118], the activation of Nrf2 after SFN treatment in healthy myoblasts [117] and muscle stem cells [115] is somewhat contradictory, as SFN treatment may delay myotube differentiation. This discrepancy may be attributed to the different cell types and growth stages being studied but identifying the effects of SFN on muscle regeneration and muscle stem cell proliferation is worthy of more comprehensive investigation.

**Table 3 antioxidants-11-00183-t003:** Effects of SFN on skeletal muscle in vivo and in vitro.

Target	Experimental Setting	SFN Treatment	Findings	References
Attenuation of muscle atrophy via regulation of Akt/FoxO1	In vitro C2C12 myotubes	Dexamethasone (5 μM) and SFN (5 μM) for 24 h	↑ Akt phosphorylation	[40]
			↑ protein synthesis	
			↑ MyoD	
			↓ atrogin-1 via FoxO1 signaling	
			↓ myocyte viability and no myotube death at 20 μM of SFN	
Nrf2/ARE signaling pathway vs. menadione-induced oxidative stress	In vitro Dystrophin knock-down muscle cells, C2 DysKD myotube	Myotubes were pre-treated with SFN (5 μM) for 5 h prior to menadione (20 μM) exposure	↑ Nrf2 translocation into nucleus	[39]
			↑ phosphorylation of Akt and Nrf2	
			↑ expression of total Akt	
Repression of myostatin and myostatin related signaling pathway	In vitro porcine satellite cells	Myoblasts were treated with 5, 10 or 15 μM SFN for 48 h	↓ myostatin gene expression	[115]
			↑ Smad7, Smurf1 gene expression at all doses	
			↓ myostatin signaling pathway	
			↓ MyoD	
			↓ cell proliferation at 15 μM	
			↓ Caspase 3 and 9 activities at 10 μM	
			↑ cell viability at 5 μM	
Modulation of CX3CL1/CX3CR1 axis and inflammation against palmitic acid-induced cell injury	In vitro C2C12 myotubes	C2C12 cells were pre-treated with SFN (5 µM) before exposure to 750 µM palmitate for 24 h	↑ cell viability	[118]
			↓ IL-6, TNF-α	
			↑ expression of Nrf2, HMOX1	
			↓ palmitic acid-induced ROS level	
Attenuation of serum starvation-induced and oxidative stress-induced muscle atrophy via Nrf2 activation	In vitro C2C12 myotubes	C2C12 cells were pre-treated for 3 h with SFN (5 µM) before exposure to 20 µM menadione, or culturing without serum for 3 h	In both conditions: Prevention against muscle damage	[116]
			↓ dichlorofluorescin diacetate (DCFDA) intensity, a general ROS indicator	
Attenuation of dystrophic pathology and muscle inflammation via Nrf2 pathway	In vivo 4-week-old *mdx* mice	2 mg/kg/day SFN via oral gavage for either 4 or 8 weeks	↑ body weight, ↓ gastric and myocardial muscle hypertrophy	[109,119]
			↓ creatine kinase (CK) and lactate dehydrogenase (LDH) levels	
			Activated Nrf2/ARE pathway	
Attenuation of muscle fibrosis via Nrf2 pathway	In vivo 3-month-old *mdx* mice	2 mg/kg/day SFN via oral gavage for 3 months	↓ CK and LDH levels	[43]
			↓ MDA, GSH/oxidized glutathione (GSSG) ratio	
			Anti-fibrosis function in liver and lungs	
			↓ p-smad2/3 = suppress profibrogenic gene	
			↓ IL-6, CD45, TNF-α = ↓ inflammatory	
			Inhibition of TGF-β/Smad signaling	
Enhancement of exercise endurance capacity via Nrf2 activation	In vivo 13-week-old male mice: Nrf2^+/+^ or Nrf2^−/−^	25 mg/kg SFN via i.p. injection 4 times in 3 days	↑ HMOX1, NQO1, gamma-glutamylcysteine synthetase (γ-GCS), and catalase	[120]
			↓ CK and LDH levels	
			↓ GSH/GSSG ratio, thiobarbituric acid reactive substances (TBARS)	
			Activated Nrf2 to ↓ muscle fatigue	
Protection against muscle damage induced by exhaustive exercise	In vivo 4-month-old male Wistar rats	25 mg/kg/day SFN via i.p. injection for 3 days prior to intensive exercise	↓ CK and LDH levels	[121]
			↑ glutathione reductase (GR), GST, NQO1	
			Activated Nrf2/ARE pathway	
Prevention of age-associated muscle dysfunction via Nrf2 signaling pathway	In vivo 21/22-month-old male C57BL/6 mice	442.5 mg/kg D, L-SFN supplemented diet for 12 weeks	↑ survival, no body weight change	[122]
			↑ exercise capacity	
			↑ MyoD, paired box 7 in satellite cell-derived myoblasts isolated from extensor digitorum longus muscles of lower hindlimbs	
			↓ myostatin, 8OHdG (oxidation marker), apoptosis	
			↑ Catalase, SOD1, Gpx1, GSTA4 and Nrf2 mRNA	
			Activated Nrf2/ARE pathway	
Protection against type 2 diabetes related muscle dysfunction	In vivo 20-week-old *db/db* type 2 diabatic mice	daily 0.5 mg/kg SFN via i.p. injection for one month	↑ grip strength	[123]
			↑ lean mass, ↓ fat mass	
			Restored muscle fiber structure	
			↓P65, TNF-α, plasminogen activator inhibitor-1, TGF-β1, Caspase 3 and Caspase 8	
			Activated Nrf2/ARE pathway	

↓—decreased and ↑—increased.

In animal models in vivo, SFN pre-treatment (25 mg/kg/day for three days) reduced muscle damage in rats caused by exhaustive exercise, evidenced by decreased levels of CK and LDH [121], and increased expression and activity of GST, GR and NQO1 in vastus lateralis muscles, linked to activation of Nrf2 signaling [121]. Similarly, mice treated with SFN (25 mg/kg) four times prior to exercise had improved exercise endurance capacity by overcoming oxidative stress-induced muscle damage [120]. Pre-treatment with SFN enhanced gene expression of antioxidants such as HMOX1, NQO1, γ-GCS and catalase in mouse gastrocnemius muscles, and protected against oxidative stress damage by reducing oxidative biomarkers including CK, LDH, GSH/GSSG ratio and TBARS [120]. Whether SFN exerts its effects directly on skeletal muscle, or via systematic protection against oxidative stress, remains unresolved.

SFN has been reported to improve skeletal muscle pathology in mouse models of Duchenne muscular dystrophy (DMD) [109,119], type 2 diabetes [123] and sarcopenia [122]. Both short-term (four weeks) and long-term (three months) treatment with SFN improved the dystrophic muscle pathology by reducing oxidative stress and inflammation [43,109]. SFN (0.5 mg/kg/day for one month) also attenuated muscle dysfunction related to type 2 diabetes, by increasing muscle mass and improving muscle function and structure in vivo [123]. A SFN-supplemented diet for 12 weeks, prolonged survival, increased muscle function and improved muscle regeneration in aged mice [122]. While it is difficult to compare the effects of SFN across studies due to differences in routes of administration, the dose and treatment duration (Table 3), taken together, these data support SFN as a potential treatment for cancer cachexia because of its anti-cancer effects and properties that confer muscle protection.

### 6.3. Honokiol and Magnolol

Honokiol and its isomer magnolol are two major components in Magnolia officinalis, a traditional Chinese herb [124]. Honokiol and magnolol are polyphenolic compounds with antioxidant and anti-inflammatory properties [125,126] shown to reduce LPS-induced inflammation and nitric oxide expression via inhibition of NF-κB signaling [127]. Magnolol has been shown to overcome hydrogen peroxide-induced oxidative stress in the human lens epithelial cell culture [125]. Magnolol can also activate detoxifying and antioxidative enzymes, including GST, in healthy ICR mice, further confirming magnolol’s antioxidative actions [128]. While honokiol and magnolol are less well studied antioxidants, they have been proposed to have anti-cancer functions, with in vitro and in vivo studies revealing inhibition of tumor-cell proliferation and growth, and induction of apoptosis [129].

In addition to these proposed anti-cancer functions, both honokiol and magnolol have reported beneficial functions in skeletal muscle. Oral administration of honokiol (1 mg/kg/day) for five consecutive days reduced muscle damage induced by eccentric exercise in adult male rats, reducing the production of proinflammatory cytokines, cyclooxygenase-2 and inducible nitric oxide synthase, via inhibition of NF-κB signaling [130]. Magnolol attenuated cancer-induced muscle wasting of C2C12 myotubes in vitro after exposure to C-26 cancer cell conditioned media. Myotube atrophy was attenuated through increased protein synthesis and reduced protein degradation in a dose-dependent manner, decreasing cancer-induced myostatin expression and subsequent downstream signaling [131]. In that study, magnolol was administered along with C-26 cancer cell conditioned media, at the beginning of differentiation, making it difficult to discern whether the treatment acted during differentiation or maturation. Magnolol has been shown to have beneficial effects on skeletal muscle in a mouse model of bladder cancer [132], with magnolol supplementation (1 mg/kg/day for three weeks via intraperitoneal injection) and chemotherapy significantly reducing cancer-induced weight loss and chemotherapy-induced anorexia [132]. While the mass of the gastrocnemius muscle was not increased compared with the untreated tumor-bearing group, magnolol supplementation significantly reduced proteasomal activity, as well as myostatin and MuRF1 expression via IGF-1-regulated signaling cascades [132]. Compared with the chemotherapy-only group, magnolol supplementation reduced production of proinflammatory cytokines, including TNF-α, IL-6 and IL-1β, via suppression of NF-κB activation [132]. Therefore, magnolol may complement chemotherapy to achieve a greater benefit, highlighting the need for combination approaches to treat cancer cachexia. While there is evidence that honokiol and magnolol have anti-cancer and muscle protecting properties, these studies are preliminary and require confirmation.

### 6.4. Pomegranate Extract

Pomegranate extract (PE) contains high polyphenol contents derived from pomegranate juice, peel and seed oil of *Punica granatum* fruit, including pomegranate ellagitannins (punicalagin), ellagic acid and gallic acid [133,134,135]. Among these polyphenols, punicalagin is the most abundant bioactive ellagitannin of pomegranate fruit [136]. PE has been widely discussed in cancer research as a potential nontoxic chemo-preventive dietary agent [133] with its anti-cancer properties demonstrated through inhibiting proliferation of MCF-7 breast cancer cells in vitro [137]. In addition, PE has been suggested as a promising dietary treatment for the prevention of androgen-independent prostate cancer both in vitro and in vivo [133]. However, clinical trials applying long-term PE supplementation for localized prostate cancer patents failed to demonstrate detectable improvement after the 18-month treatment [138]. While PE has anti-cancer and anti-inflammatory effects in lung, skin and colon cancer in preclinical in vitro and in vivo studies, there are limited clinical data and optimal dose and treatment duration have yet to be determined.

In addition to the potential anti-cancer effects of PE, studies have also suggested its ability to protect skeletal muscle. Oral supplementation with a commercial punicalagin-rich PE (Pomanox-P30) for six weeks alleviated TNF-α-induced activation of NF-κB signaling cascade, reduced cytokine induction and protected against skeletal muscle mass loss in mice with acute TNF-α-induced inflammation [139]. PE supplementation also reduced TNF-α-induced oxidative stress, as evidenced by reduced mRNA expression of *nox2* and *nox4*, indicating reduced oxidation with PE treatment [139]. The study also suggested that PE pre-treatment maintained activity of the Akt/mTORC1 pathway, and limited activation of the ubiquitin proteasome pathway, indicating favoring of protein synthesis over degradation [139]. Therefore, while limited data are currently available, PE is proposed to have protective effects against the loss of muscle mass. Further studies need to confirm these proposed benefits of PE on skeletal muscle. PE has gained attention as a food supplement in sport science, with high-polyphenol containing pomegranate juice given twice daily to participants for seven days prior to intense eccentric exercise, found to enhance strength and delay muscle soreness [140]. Several human trials have shown promising outcomes for PE in different exercise modalities by improving performance, enhancing power output and vascular oxygen content and accelerating muscle recovery [141]. However, the mechanisms underlying these benefits of PE on skeletal muscle remain to be determined and studies should investigate these effects at the cellular level to confirm potential therapeutic benefit. Lastly, PE has also been shown to alleviate oxidative stress and inflammation induced by the chemotherapeutic 5-Fluorouracil (5-FU) in human immortal keratinocyte cells, suggesting the potential for PE to protect against the off-target effects of 5-FU chemotherapy [142].

### 6.5. Ellagic Acid and Urolithin A, B

Ellagic acid (EA) is a naturally occurring phenolic constituent that is a subgroup of ellagitannins found in many fruits and plants, including grapes, nuts, berries, green tea and pomegranates [143]. EA has significant chemo-preventive effects against different types of cancer, including breast, prostate, colon and skin cancer, and elicits anti-inflammatory and antioxidative effects in vitro and in vivo [143]. Pre-treatment with EA (100 mg/kg) daily for one week significantly reduced skeletal muscle damage induced by ischemia/reperfusion (I/R) and ameliorated I/R-induced oxidative stress by decreasing the level of MDA and inducing antioxidative enzymes, including SOD, catalase and GSH-Px in gastrocnemius muscles of Sprague–Dawley rats [144]. Similarly, EA attenuated the loss of muscle mass in carbon tetrachloride-induced muscle wasting [145]. In Wistar rats, intraperitoneal injection of EA (10 mg/kg, five times per week for eight weeks) decreased carbon tetrachloride-induced muscle damage and oxidative stress by increasing expression of caspase-3 and Nrf2 while decreasing B-cell lymphoma 2 (bcl-2), NF-κB, TNF-α, MDA and cyclooxygenase-2 (COX-2) expression [145]. Based on these preliminary findings, EA may have potential to attenuate cancer cachexia. However, discrepancies in methodology, including variations in dose, route of administration and duration of treatment, limit rigorous comparisons between these studies, and highlight the need for better controlled studies.

Urolithin A and B are the metabolites of EA or ellagitannin, with slight variations in phenolic hydroxylation patterns [146]. Urolithins are normally produced by the microflora in the gastrointestinal tract after digestion of fruits containing high level of EA or ellagitannin [146,147]. Due to the low bioavailability and aqueous solubility of EA, it is likely that these colonic metabolites are the bioactive compounds responsible for the reported anti-inflammatory, antioxidative and anti-cancer effects of EA [148]. Urolithin A and B have reported anti-cancer effects [149] with additional benefits in skeletal muscle. Pre-treatment with urolithin A (15 μM) for 24 h attenuated TNF-α-induced inflammation and prevented activation of NF-κB signaling to protect wasting of C2C12 myotubes in vitro [139]. Urolithin A is a first-in-class activator of mitophagy [150]. Since reduced mitophagy is one of the hallmarks of the pathology in DMD, urolithin A has been trialed in studies using the well-characterized *mdx* mouse model of DMD [150]. Dietary supplementation with urolithin A (50 mg/kg) for 10 weeks restored disrupted mitophagy and mitochondrial respiratory capacity in *mdx* mice [150]. In addition, urolithin A treatment enhanced myocellular quality by increasing expression of α-dystrobrevin and β-dystroglycan, the structural proteins required to restore muscle morphology in *mdx* mice, and it reduced the percentage of centrally nucleated fibers (a hallmark of regeneration), and increased muscle cross sectional area [150]. Furthermore, 24 h treatment of urolithin B (15 μM) enhanced the growth and differentiation of C2C12 myotubes by increasing protein synthesis and attenuating protein degradation [147]. Administration of urolithin B (10 μg/day) via mini-osmotic pumps in C57BL/6 mice for three weeks, increased body mass and increased the mass of tibialis anterior, soleus and quadriceps muscles [147]. Furthermore, urolithin B (10 μg/day for seven days) attenuated denervation-induced muscle atrophy through suppression of FoxO1 and FoxO3, MAFbx, MuRF1 and myostatin expression [147]. Overall, ellagic acid, urolithin A and urolithin B are potent antioxidants with anti-cancer and muscle-protective effects. Their therapeutic potential for cancer cachexia requires further investigation.

### 6.6. Other Polyphenols

In addition to the polyphenols already discussed, evidence has emerged over the past 5–10 years of other polyphenol compounds with the potential to protect against cancer cachexia. Hesperidin is a natural occurring flavonoid with significant antioxidative properties, most abundant in citrus fruits, such as oranges, tangerines and lemons [151]. It has demonstrated anti-cancer properties through attenuating the growth of breast, lung and colon cancer cells in vitro and in vivo, via induction of apoptosis, regulating cell cycle and reducing oxidative stress [151]. The combination of hesperidin (50 mg/kg/day, orally) and doxorubicin for 16 days significantly reduced tumor size and mass, while overcoming chemo-resistance to doxorubicin in Ehrlich ascitic tumor-bearing mice [152]. This highlights a potential role for hesperidin as a combination therapy for treating cancer. In addition to its anti-cancer functions, pre-treatment with hesperidin (100 mg/kg/day, orally), one week prior to ischemia/reperfusion (I/R) injury, protected skeletal muscle from subsequent I/R injury-induced oxidative stress through reductions in MDA and increased SOD expression in gastrocnemius muscles of Sprague–Dawley rats [144]. Further investigation is required to determine its mechanism of action in skeletal muscle and the potential to ameliorate cancer-induced muscle wasting.

Pterostilbene is a bioactive polyphenol, found in grapes, wine and berries, which shares similar chemical properties with resveratrol but has greater in vivo bioavailability, suggesting better translational potential [153]. Similar to resveratrol, pterostilbene has chemo-protective and chemo-preventive properties, with demonstrated effects in various cancers, including breast, colon, prostate and liver, through activation of Nrf2 signaling and antioxidant activities, inhibition of STAT3 activity and induction of apoptosis [154]. In vitro, pterostilbene (100 μM) inhibited cancer cell growth over time (24–72 h) in chemo-resistant T24 cells, indicating pterostilbene may have potential in combination therapies to increase sensitivity of chemotherapy in cancer [155]. In addition, oral pterostilbene supplementation (50 mg/kg/day) to Sprague–Dawley rats for four weeks promoted muscle adaptation and enhanced endurance capacity during exercise training by promoting slow-twitch fiber formation, increasing angiogenic factor expression and improving mitochondrial function [156]. In a mouse model of collagen-VI-deficient of myopathy, pterostilbene (90 mg/kg/day, oral gavage for five days) restored autophagy in autophagy-deficient muscle, counteracting the major pathogenic marker of collagen-VI-related myopathies [157]. Lastly, pre-treatment with pterostilbene (2.5, 5 or 10 mg/kg/day, intraperitoneal injection) to Sprague–Dawley rats one week prior to I/R injury inhibited subsequent I/R injury-induced oxidative stress in a dose-dependent manner by reducing apoptosis and activating SIRT1-FoxO1/p53 signaling in skeletal muscle [153]. Based on the beneficial effects of pterostilbene on skeletal muscle and its anti-cancer properties, future studies should investigate the potential for pterostilbene to attenuate cancer-induced muscle wasting.

## 7. Dual-Function of Antioxidants in Cancer Cachexia

Targeting oxidative stress has significant promise to treat cancer cachexia but conflicting findings question the safety of antioxidant therapy in some cancer populations. Administration of an antioxidant cocktail enriched in EGCG, curcumin and vitamin C, had no benefit in cancer cachexia and instead enhanced weight loss in tumor-bearing mice, resulting in premature death [158]. Individually, EGCG [34] and curcumin [28] have been shown to counteract tumor growth and ameliorate cancer cachexia in vivo, but their combination failed to improve skeletal muscle atrophy and survival in the C-26 mouse model of cancer cachexia, likely due to the failure to reduce plasma TNF-α levels, systematic oxidative stress and ubiquitin-proteasome system activation [158]. In addition to the controversy surrounding the efficacy of antioxidants for cancer prevention, the potential for antioxidants to treat cancer cachexia remains equivocal. The dual effect of ROS may be responsible for this ambiguity since they can both promote proliferation and induce cancer cell death [159]. Cancer cells can influence levels of ROS and antioxidants to achieve a steady ROS state, avoiding oxidative-stress-induced cell apoptosis [159,160]. Therefore, antioxidant supplements could be either beneficial or deleterious depending on the cancer conditions. Reducing oxidative stress might accelerate or attenuate tumor growth depending on the tumor type and location [158].

The benefits of antioxidant supplements for treating cancer cachexia in humans remains inconclusive. Among different types of cancer, the benefits of antioxidants were limited to cachectic patients with reduced antioxidant activity and high level of ROS in the blood [161]. These findings suggest that antioxidant therapies may be beneficial for attenuating cachexia in patients with antioxidant deficiencies [75] but potentially harmful for patients with a normal redox balance [75,161]. Therefore, the therapeutic application of antioxidants for cancer cachexia remains questionable because of these competing effects on ROS regulation. Given the contradictory data reported to date, it is important to better understand the mechanisms underlying the action of antioxidants and their regulation of tumor growth and muscle mass, particularly for different cancer types and disease stages. Furthermore, it is crucial that optimal models for specific cancers be identified to facilitate the most appropriate pre-clinical studies that can inform clinical translation.

## 8. Conclusions

This review highlighted the potential for dietary antioxidants to attenuate cancer-induced muscle wasting, identifying those able to attenuate cancer-induced muscle wasting, some with specific anti-cancer effects and others with potential muscle protective functions that have yet to be tested in the context of cancer cachexia. Considering the balance between the positive and negative effects of antioxidants that have been identified so far, therapeutic application of antioxidants for cancer cachexia requires closer examination. Since cancer cachexia is a multifactorial syndrome, combinatorial treatments will likely be necessary. As antioxidants are commonly used as dietary supplements, their role and application in these multimodal treatments is worthy of further investigation.

## Figures and Tables

**Figure 1 antioxidants-11-00183-f001:**
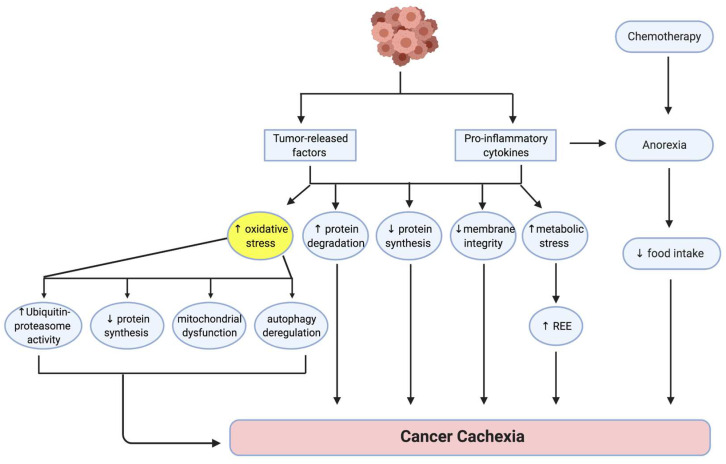
Pathogenesis of cancer cachexia. Numerous factors influenced by host cytokines and tumor-released factors result in an imbalance between protein degradation and protein synthesis that ultimately leads to muscle wasting and weakness [11,12]. Membrane integrity is compromised because of potential dysfunction of the dystrophin-glycoprotein complex [13]. Metabolic dysregulation leads to an increased resting energy expenditure (REE) and this contributes to tumor progression and nutritional deficiencies [9]. Increased oxidative stress is a key contributor to cancer cachexia, contributing to mechanisms that favor protein breakdown over protein synthesis through increased ubiquitin proteasome activity, mitochondrial dysfunction and dysregulation of autophagy. Chemotherapy can contribute to the pathology of cancer cachexia by inducing anorexia [14]. Adapted from [10]. Figure created using Biorender.com, accessed on 7 November 2021.

**Figure 2 antioxidants-11-00183-f002:**
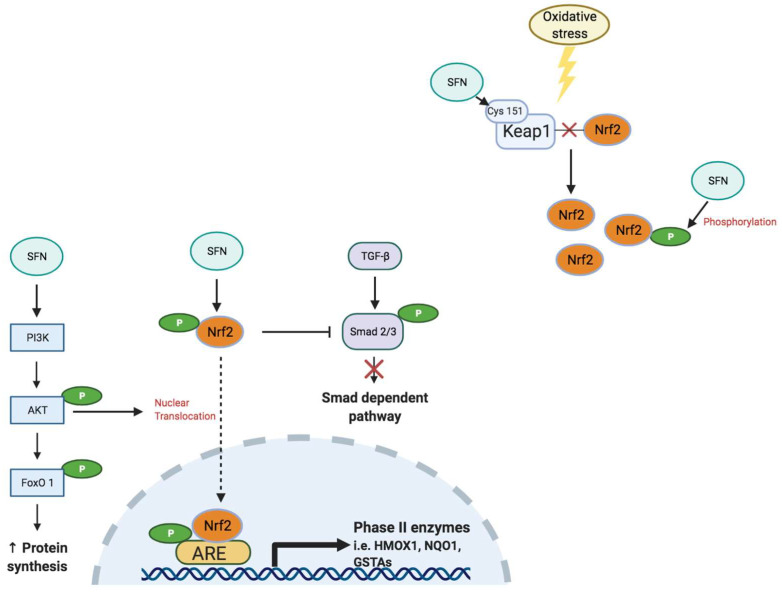
Sulforaphane (SFN) activates the Nrf2 signaling pathway. Under normal conditions, Nrf2 localizes to the cytoplasm, where its activity is suppressed by Keap1. When cells experience extensive oxidative stress, Nrf2 is released from the Keap1 complex and translocates into the nucleus, where it promotes antioxidant and detoxifying gene expression via the antioxidant response element (ARE) complex [39]. SFN regulates activation of Nrf2 and increases the rate of nuclear translocation of Nrf2 [41]. SFN activates Nrf2 via direct modification of critical Keap1 cysteines, such as Cys 151 [42]. SFN can also de-methylate the promoter region of Nrf2 and accelerate Nrf2 protein synthesis [36,42]. Phosphorylated Nrf2 also inhibits phosphorylation of Smad2/3 in the transforming growth factor-β (TGF-β) signaling pathway linked to an attenuation of tissue fibrosis progression [43]. In addition, SFN can increase protein synthesis and decrease protein degradation via activation of the protein kinase B (Akt)/Forkhead box O1 (FoxO1) signaling pathway [40]. Figure created using Biorender.com, accessed on 7 November 2021.

**Table 2 antioxidants-11-00183-t002:** Effects of Ursolic acid on skeletal muscle in vivo and in vitro.

Target	Experimental Setting	UA treatments	Findings	References
Regulating muscle mass and overcoming different muscle wasting in various conditions	In vivo 6–8-week-old male C57BL/6 mice	24 h fasting: i.p. injection of 200 mg/kg given at 0 and 12 h time points	Reduced fasting-induced muscle atrophy:	[97]
			↑ lower limb muscle mass by 7% ± 2%	
			↓ atrogin-1 and MuRF1 mRNA expression	
		Denervation: i.p. injection of 200 mg/kg, twice daily for 7 days	Decreased denervation-induced muscle loss:	
			↑ muscle fiber size (hindlimb muscles)	
		Hypertrophy test: Chow containing 0.27% UA for 5 weeks	Induced muscle hypertrophy:	
			↑ muscle mass, fiber size and grip strength	
			↑ IGF-1 mRNA expression	
Sustaining resistance exercise-induced mTORC1 activity	In vivo 10-week-old male Sprague–Dawley rats	250 mg/kg, i.p. injection right after exercise	Sustained phosphorylation of p70S6K	[100]
			↑ Akt Thr308 phosphorylation	
			Failed to increase muscle IGF-I concentrations with UA treatment alone	
Increasing muscle mass and function	In vivo 8-week-old male C57BL/6 mice	High fat diet supplemented with 0.14% UA for 6 weeks	↑ Akt phosphorylation	[99]
			↑ muscle mass	
			↑ slow and fast muscle fiber size	
			↑ grip strength	
			↑ food intake, energy expenditure	
Promoting muscle regeneration	In vitro Skeletal muscle satellite cells isolated from 10-day-old C57BL/6 mice	In vitro 10 μM treatment for 10 days	In vitro	[101,102]
			↑ Pax7 expression	
			↑ sirtuin 1 (SIRT1), Peroxisome proliferator-activated receptor-gamma coactivator-1α (PGC-1α) expression	
	In vivo 10-month-old male C57BL/6 mice	In vivo i.p. injection of 200 mg/kg, twice daily for 7 days	In vivo	
			↓ body weight	
			Change muscle composition to be more oxidative	
			↑ myoglobin expression	
			↓ cellular energy status (ATP, ADP)	
			↑ fiber generation	
Attenuating hypobaric hypoxia-induced skeletal muscle wasting via Akt signaling pathway	In vivo Male Sprague–Dawley rats	20 mg/kg/day via oral gavage for 3 days	↓ ROS level, protein oxidation	[98]
			↑ antioxidative enzymes: GPx, GR, SOD1, SOD2 and catalase	
			↑ glutathione (GSH) activation	
			↓ caspases 3	
			↓ IL-1β, IL-10, IL-4, TNF-α	
			↓ weight loss	
			↑ muscle protein contents	
			↑ grip strength	
			Enhanced Akt phosphorylation, IGF-1 protein expression, p70S6K	

↓—decreased and ↑—increased.

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
