# Peer review of "Role for Plant-Derived Antioxidants in Attenuating Cancer Cachexia"

_antioxidants, 2022, doi:10.3390/antiox11020183_

Round 1

Reviewer 1 Report

It is a well and clearly written paper, but there are some recommendations I would like to address. Firstly, please rephrase the following lines:  12-13, 26-27, 57, 206.

I suggest rethinking the title as the authors specified that dietary antioxidants can only “attenuate cancer-induced muscle wasting”, not treating it, as there are also positive and negative effects of antioxidants.

I would find it interesting if the authors approach the involvement of antioxidants in attenuating the negative effects of chemotherapy because it is another important factor in cancer cachexia induction.

Regarding section 5, I consider that the information about polyphenols is very few. Why did you choose to talk about polyphenols as emerging agents for cancer cachexia? Moreover, there are more examples of polyphenols with anticachectic potential, for example, rutin. (see: Gil da Costa, et all. HPV16 Induces a Wasting Syndrome in Transgenic Mice: Amelioration by Dietary Polyphenols via NF-ΚB Inhibition. Life Sci. 2017, 169,11–19. https://doi.org/10.1016/J.LFS.2016.10.031.)

Please check and argue your choosing if consider those mentioned as the most important, or else, increase the spectrum of information delivered.

Reviewer 2 Report

The paper submitted for review presents knowledge about plant-delivered antioxidants in cachexia, mainly polyphenols. Unfortunately, the manuscript omitted information about such essential compounds as gallic or ellagic acid. That proves that the authors treat the subject quite superficially. They did not consider knowledge about, for example, Urolithin B or quercetin. For this reason, I strongly recommend the authors once again review the literature on the subject and consider a broader spectrum of plant-origin compounds in their rewritten paper. In its present form, it is incomplete and thus misleads the readers, probably not intentionally, and severely limits their knowledge of the subject. Therefore, it does not meet the basic principles and functions of review papers.

Round 2

Reviewer 2 Report

Authors enriched the paper according to my suggestion so I accept the manuscript in the present form